# Could *SLC26A7* Be a Promising Marker for Preoperative Diagnosis of High-Grade Papillary Thyroid Carcinoma?

**DOI:** 10.3390/diagnostics14232652

**Published:** 2024-11-25

**Authors:** Sergei E. Titov, Evgeniya S. Kozorezova, Sergei A. Lukyanov, Sergei V. Sergiyko, Pavel S. Demenkov, Yulia A. Veryaskina, Sergey L. Vorobyev, Ilya V. Sleptsov, Roman A. Chernikov, Natalia I. Timofeeva, Svetlana V. Barashkova, Elena L. Lushnikova, Anna A. Uspenskaya, Anna V. Zolotoukho, Olga V. Romanova, Igor F. Zhimulev

**Affiliations:** 1Department of the Structure and Function of Chromosomes, Institute of Molecular and Cellular Biology, Siberian Branch of the Russian Academy of Sciences, Novosibirsk 630090, Russia; microrna@inbox.ru (Y.A.V.); zhimulev@mcb.nsc.ru (I.F.Z.); 2PCR Laboratory, AO Vector-Best, Novosibirsk 630117, Russia; 3Department of Natural Sciences, Novosibirsk State University, Novosibirsk 630090, Russia; demps@math.nsc.ru; 4National Center of Clinical Morphological Diagnostics, Saint Petersburg 192283, Russia; kozorezovaes@yandex.ru (E.S.K.); slvorob@gmail.com (S.L.V.); svbarr@yandex.ru (S.V.B.); 5Department of General and Pediatric Surgery, South Ural State Medical University, Chelyabinsk 454092, Russia; 111lll@mail.ru (S.A.L.); ssv_1964@mail.ru (S.V.S.); 6Institute of Cytology and Genetics, Siberian Branch of the Russian Academy of Sciences, Novosibirsk 630090, Russia; 7Department of Endocrine Surgery, Saint Petersburg State University Hospital, Saint Petersburg 199034, Russia; newsurgery@yandex.ru (I.V.S.); yaddd@yandex.ru (R.A.C.); natalytim@mail.ru (N.I.T.); uspenskaya_anna@mail.ru (A.A.U.); a.zolotoukho@gmail.com (A.V.Z.); 8Department of Molecular Pathology, Federal Research Center of Fundamental and Translational Medicine, Novosibirsk 630117, Russia; pathol@inbox.ru; 9N.N. Blokhin National Medical Research Center of Oncology, Moscow 115478, Russia; olga-romanova1995@mail.ru

**Keywords:** papillary thyroid carcinoma, high-grade carcinoma, preoperative diagnosis, molecular marker, *SLC26A7*

## Abstract

Background/Objectives: A modern classification distinguishes between two nosological entities posing an intermediate risk between differentiated and anaplastic carcinoma: poorly differentiated thyroid carcinoma and differentiated high-grade thyroid carcinoma. There are currently few studies searching for the preoperative molecular genetic markers of high-grade papillary thyroid carcinoma (PTC HG), primarily because of a recent WHO reclassification and singling out of a separate entity: high-grade follicular cell-derived nonanaplastic thyroid carcinoma. Therefore, this work was aimed at identifying PTC HG-specific microRNAs and mRNAs that reliably distinguish them from differentiated papillary thyroid carcinoma in preoperative cytology specimens (fine-needle aspiration biopsies). Methods: A molecular genetic profile (expression levels of 14 genes and eight microRNAs) was studied in 110 cytology specimens from patients with PTC: 13 PTCs HG and 97 PTCs without features of HG. Results: Of the examined eight microRNAs and 14 genes, significant differences in the expression levels between the PTC and PTC HG groups were revealed for genes *SLC26A7*, *TFF3*, and *TPO*. Only one gene (*SLC26A7*) proved to be crucial for detecting PTC HG. It showed the largest area under the ROC curve (0.816) in differentiation between the PTC and PTC HG groups and was the key element of the decision tree by ensuring 54% sensitivity and 87.6% specificity. Conclusions: Early preoperative diagnosis of PTC HG in patients with early stages of this cancer type will allow clinicians to modify a treatment strategy toward a larger surgery volume and lymph node dissection and may provide indications for subsequent radioactive iodine therapy.

## 1. Introduction

The *WHO Classification of Endocrine Tumors* (the 5th Edition, 2022) distinguishes between two subtypes of high-grade follicular cell-derived nonanaplastic thyroid carcinoma, posing a risk that is intermediate between differentiated and anaplastic thyroid carcinoma: poorly differentiated thyroid carcinoma (PDTC) and differentiated high-grade thyroid carcinoma (DHGTC) [1]. DHGTCs are high-grade epithelium-derived carcinomas that retain their structural follicular/papillary differentiation and cytological nuclear features, typical of papillary carcinoma. DHGTC is characterized by invasive growth, a necrotic component, and an elevated mitotic index while retaining signs of differentiation, which is not typical of anaplastic carcinoma [2].

The need to introduce a new nosological entity—differentiated high-grade thyroid carcinoma—has arisen because papillary thyroid carcinoma (PTC, which represents the majority of differentiated thyroid carcinoma [DTC] cases) most often progresses indolently and does not pose a risk of death. However, researchers also see aggressive subtypes of PTC, which progress less favorably and have a worse prognosis. Moreover, in some countries, such as the United States, mortality from thyroid cancer has increased by 1.1% per year from 1994 to 2013, whereas during the same period, the annual increase in the incidence of (and mortality from) PTC is 1.7% [3].

Most DHGTCs not only have cytoarchitectonic features of papillary carcinoma but can also carry the BRAF V600E mutation [4,5] while remaining prone to regional metastasis to cervical lymph nodes [5]. The term “papillary thyroid carcinoma, high grade” (PTC HG) has been proposed for this type of tumor. These tumors usually develop in adults, are frequently metastatic (20–50%), characterized by substantial local invasion, and are often resistant to radioactive iodine therapy [6,7]. Postoperative treatment of this group of patients can involve external beam radiation therapy and/or targeted chemotherapy with tyrosine kinase inhibitors. DHGTC patients have a similar survival rate (56% 10-year survival rate) [5], although recurrence-free survival (without signs of disease) of DHGTC patients can be lower compared to that of PTC patients [7]. The unfavorable prognosis is also related to the fact that patients often receive a diagnosis of DHGTC at late stages. Therefore, an early and accurate diagnosis of this cancer is crucially important for initiating effective treatment [8].

Fine-needle aspiration biopsy (FNA) is a minimally invasive and highly effective method of PTC detection, but the specificity to HG signs is modest. PTC HG is cytologically verified regarding papillary carcinoma or a tumor suspected of papillary carcinoma (Bethesda V/VI); most often, there are no sufficient features to expect high-grade carcinoma. PTC HG can be suspected based on clinical data, such as the large size and rapid growth of the tumor, as well as the presence of regional and distant metastases [9].

DHGTC (and PTC HG in particular) typically develops from a pre-existing well-differentiated thyroid carcinoma as a result of genetic rearrangements; therefore, DHGTC (PTC HG) is caused by the known mutations in MAPK and PI3K–AKT signaling pathways [10,11]; accordingly, it is rather difficult to differentiate between papillary carcinoma and PTC HG at the molecular level. Meanwhile, some available data are indicative of an association of high-grade carcinomas with additional mutations in *TP53* and *TERT* promoters and/or dysregulated expression of certain microRNAs (miRNAs, miRs) [12,13], although no specific genetic markers have been identified yet for this group of patients.

Various molecular genetic markers (miRNA and messenger RNA [mRNA]) are currently successfully employed for diagnosing differentiated forms of thyroid cancer in such genetic testing panels as the ThyGenX/ThyraMIR, Afirma Genomic Sequencing Classifier, and mir-THYpe [14,15,16]. These panels can detect pituitary cancer with high accuracy in patients with uncertain cytological findings (Bethesda III, IV, and V). A panel developed earlier [17,18] also allows one to improve the preoperative stratification of thyroid tumors into benign and malignant ones in FNA smears, as well as to distinguish thyroid carcinoma types such as oncocytic, papillary, and medullary with high accuracy. Nonetheless, this panel has proved to be insufficiently informative for verifying poorly differentiated thyroid carcinomas [19,20].

In comparison with anaplastic thyroid cancer, the probability of detecting a high-grade tumor of small size for the first time is much higher [21]. If the tumor size is less than 4.0 cm, then a surgeon may decide to perform a hemithyroidectomy. Because PTC HG is an aggressive tumor type, with the above treatment strategy, the patient will need a second surgical operation to the extent of removal of thyroid gland remnants and a level VI lymphadenectomy of the neck; the latter modalities are always associated with an elevated risk of complications (laryngeal nerve paresis and hypoparathyroidism) [22]. Suspicion of the high grade already at the preoperative stage will allow a surgeon to immediately choose the correct extent of the surgical intervention.

There are currently very few studies searching for the preoperative molecular genetic markers of PTC HG, primarily because of the recent WHO reclassification of endocrine tumors (2022, 5th edition) and the singling out of a separate nosological entity: high-grade follicular cell-derived nonanaplastic thyroid carcinoma. Therefore, this work was aimed at identifying PTC HG-specific miRNAs and mRNAs that reliably distinguish them from differentiated papillary thyroid carcinoma in preoperative cytology specimens collected by FNA.

## 2. Materials and Methods

Clinical material. Two institutions took part in this study: Saint Petersburg State University Hospital (St. Petersburg, Russia) and South Ural State Medical University (Chelyabinsk, Russia). All the cytology and histology tissue samples pertaining to this study were re-examined by two independent pathologists from the National Center for Clinical Morphological Diagnostics.

To search for patients with PTC HG who meet the criteria of the WHO classification (2022), 298 histological examinations of PTC specimens derived from patients who had been operated on in 2021–2023 were reassessed. After a comprehensive re-examination of the cytology and histology specimens, 110 specimens were selected; cases not meeting the inclusion criteria (see below) were excluded from the study. Of those, the criteria for PTC HG were met in 13 cases: features of differentiated papillary carcinoma were retained (structural and nuclear morphology); the presence of necrotic foci and an increased number of mitoses ≥ 3 × 2 mm^2^; and no anaplastic dedifferentiation. The comparison group contained 97 patients with PTC without features of HG.

The inclusion criteria were as follows: cytology and histology specimens being available; age > 18 years; a match between the ultrasonography findings for a thyroid nodule during preoperative diagnostics and the macroscopic characteristics of the thyroid nodule in the operative specimen; and >200 tumor cells in the cytology specimen. Cytology specimens on glass slides prepared by the conventional procedure (fixation and May–Grünwald–Giemsa staining with azure and eosin dyes) were included in the study.

Choosing a set of molecular markers. A set of mRNAs for the analysis was primarily selected, taking into account published findings. The list of mRNAs consisted of 14 genes: *FN1*, geminin (*GMNN*), *CDKN2A*, *TPO*, *SLC26A7*, *HMGA2*, *CPQ*, *SPATA18*, *APOE*, *DIO1*, *NIS*, *TFF3*, *TMPRSS4*, and *TSHR*. MiRNAs were chosen based on our own studies [23,24,25,26,27,28,29,30,31] and literature analysis. Eight miRNAs were subjected to experimental analysis: miR-146b-5p, miR-199b-5p, miR-221-3p, miR-223-3p, miR-31-5p, miR-375, miR-551b-3p, and miR-21-5p [32,33]. The ratio of mitochondrial DNA (mtDNA) to nuclear DNA (nDNA) served as a criterion for the presence of oncocytes in a cytology specimen [20]. Therefore, 23 molecular genetic markers were investigated in this work.

Total nucleic acid extraction. Nucleic acid extraction was conducted according to a published protocol [32] with modifications: each dried cytology specimen was washed in a 1.5 mL microcentrifuge tube with three 200 μL portions of guanidine lysis buffer (4 M guanidine thiocyanate; 25 mM sodium citrate, pH 7.0; 0.3% of sarcosyl; and 0.1% of 2-mercaptoethanol). The sample was vigorously mixed and incubated in a thermal shaker for 15 min at 65 °C. Next, an equal volume (600 μL) of isopropanol was added. The reaction solution was thoroughly mixed and kept at room temperature for 5 min. After centrifugation for 15 min at 14,000× *g*, the supernatant was discarded, and the pellet was washed with 500 μL of 70% ethanol and then with 300 μL of acetone. Finally, the RNA was dissolved in 200 μL of deionized water. If not analyzed immediately, the RNA samples were stored at –20 °C. The concentration of the isolated total RNA was measured on a NanoDrop 2000C spectrophotometer (Thermo Scientific, Waltham, MA, USA). Total RNA concentrations were in the range of 5.4–51.6 ng/μL (mean 25.2 ng/μL).

Oligonucleotide primers and probes

All oligonucleotides were synthesized at AO Vector-Best (Novosibirsk, Russia). The oligonucleotides were selected using the OligoAnalyzer™ Tool online service (https://eu.idtdna.com/calc/analyzer (accessed on 22 November 2024)). Sequences of primers and fluorescently labeled probes are presented in the Appendix A. The standard primer concentration in all the reactions was 0.5 μM, whereas the concentration of the fluorescently labeled probe was 0.25 μM.

Semiquantitative assessment of the mRNA content. This procedure was performed by real-time reverse-transcription PCR (RT-PCR) employing specific primers and fluorescently labeled probes to detect the mRNA of one of the above genes and the housekeeping gene *PGK1* (phosphoglycerate kinase) used for normalization. The RT-PCR protocol was as follows: incubation at 45 °C for 30 min; heating at 95 °C for 2 min; 50 cycles: denaturation at 94 °C for 10 s; annealing and elongation at 60 °C for 20 s. Each relative expression level was calculated by the 2^−ΔCq^ method [34].

Semiquantitative assessment of the miRNA content. Eight miRNAs were detected by real-time RT-PCR for all the types of tumors and lumps. Mature miRNAs were detected by stem-loop RT-PCR [35]. The reverse-transcription reaction and real-time PCR were carried out according to a protocol reported in ref. [32]. The reverse-transcription reaction followed by real-time PCR was carried out individually for each miRNA. Single-replicate analysis was performed on each RNA sample. The miRNA content was normalized to the geometric mean of the levels of three reference miRNAs (miR-197-3p, miR-23a-3p, and miR-29b-3p) by the 2^−ΔCq^ method.

Analysis of the ratio of mtDNA to nDNA. MtDNA and nDNA were detected by real-time PCR using the following protocol: preheating at 95 °C for 2 min; 50 cycles: denaturation at 94 °C for 10 s; annealing and elongation at 60 °C for 20 s. The ratio was determined by the 2^−ΔCq^ method.

Statistical analysis. The data were analyzed in the SPSS Statistics 23 (IBM, Armonk, NY, USA) and Excel (Microsoft, Redmond, WA, USA) software. For continuous variables, two independent samples were compared by the Mann–Whitney *U* test. The significance level was set to 0.0022 after the Bonferroni correction. Receiver operating characteristic (ROC) analysis was carried out to make it possible to use miRNA and gene expression levels to detect PTC HG. A test comparison was performed with the help of areas under the ROC curves (ROC AUCs). Tissue samples were stratified into different groups via the C4.5 decision tree algorithm [36].

## 3. Results

Among the 13 patients with PTC HG, there were 10 (77%) females and 3 (23%) males. The mean age at diagnosis was 64 years (range: 48–74 years). A tumor was found to be multifocal in three (23%) cases. A macroscopically visible extrathyroidal extension was observed in seven (53.8%) patients. Metastases to the regional lymph nodes were detected in eight (61.5%) patients; no distant metastases were found (Table 1). The median tumor size was 45 mm (interquartile range: 35–50 mm); the minimal and maximal tumor sizes were 17 and 60 mm, respectively. One patient had a stage T1b tumor; five patients had stage T2; and seven patients had a stage T3b tumor.

A comparative description of the relative expression levels of eight miRNAs, a relative mtDNA level, and the expression levels of 14 genes was performed. The results are summarized in Table 2.

No significant differences in miRNA expression or the relative mtDNA level were revealed between groups PTC and PTC HG. Among the protein-coding genes, significant (with an adjustment for multiple comparisons) differences were detected for the following:The *TPO* gene (*p* = 0.00085) encodes thyroid peroxidase, which plays an important role in thyroid hormones’ synthesis and maintenance of stable thyroid function. According to other studies, *TPO* is highly expressed, mainly in normal thyroid tissue or benign lesions, whereas in thyroid cancer, its expression is reduced or absent [37].The *SLC26A7* gene (*p* = 0.000166) codes for a transmembrane Cl^−^/HCO_3_^−^ ion transporter that controls the acid–base balance in renal tubules and the gastric epithelium and appears to perform a similar function in thyroid follicular cells. This is because an optimal ionic milieu and pH are known to be critical for enzymatic activity (of TPO, for example) during thyroid hormone biosynthesis [38]. There is evidence that decreased *SLC26A7* expression accompanies the development of anaplastic carcinoma and significantly correlates with a poor prognosis among the patients [26], whereas in PTC, *SLC26A7* downregulation has been associated with an elevated risk of metastases outside the thyroid [39].The *TFF3* gene (*p* = 0.000614) encodes a small molecular peptide that belongs to a family of small secretory molecules involved in the protection and repair of the gastrointestinal mucosa. TFF proteins participate in the maintenance and restoration of epithelial structural integrity by activating key signaling pathways for epithelial–cell migration, proliferation, and invasiveness [40]. *TFF3* expression is diminished in PTC, and patients with a lower gene expression have poorer disease-free survival but a higher immune-cell infiltration [41].

Figure 1 shows the relative expression of these genes in groups PTC and PTC HG. For comparison, Figure 1 also depicts the relative expression of the genes whose *p*-values lay in the range 0.0022 < *p* < 0.05: *GMNN* (*p* = 0.028378; its protein modulates DNA replication by binding to replication factor Cdt1, thereby altering Cdt1’s stability and activity [42]), *CDKN2A* (*p* = 0.004832; the cyclin-dependent kinase inhibitor 2A gene encodes tumor suppressor protein p16INK4a, which plays an important part in inhibition of cell cycle progression [43]), *CPQ* (*p* = 0.007873; its protein may take part in a release of hormone thyroxine; underexpression has been noted in follicular thyroid cancer [24]), *SPATA18* (*p* = 0.012699; it codes for a protein induced by p53 and causes the accumulation of lysosomal proteins in damaged mitochondria to utilize oxidized mitochondrial proteins in order to restore damaged mitochondria [44]), and *NIS* (*p* = 0.016281; it encodes a sodium–iodine symporter; in most studies, *NIS* expression is lower in thyroid carcinomas than in adenomas and normal adjacent tissue [45]).

Of note, an elevated relative expression level in the PTC HG group was detected only for the *CDKN2A* (*INK4a*) gene; it codes for cyclin-dependent kinase inhibitor 2A, which acts as a negative regulator of normal cell proliferation and a tumor suppressor. Expression of other genes in the PTC HG group proved to be downregulated.

It is also worth mentioning that there is evidence that in thyroid cancer, the expression of the *SPATA18* gene is low, which leads to an increase in the number of abnormal mitochondria, in the level of reactive oxygen species, and in the mtDNA/nDNA ratio and to acidification of the cytoplasm [44]. In our work, PTC HG tissue samples featured a reduced level of *SPATA18* expression as compared to PTC without signs of HG and, at the same time, a slightly elevated mtDNA/nDNA ratio, albeit without statistical significance (*p* = 0.206242). The mtDNA/nDNA ratios in PTC and PTC HG are presented in Figure 1.

The data on the relative expression of these genes were used to calculate the ROC AUCs for assessing the ability of individual molecular markers to distinguish groups PTC and PTC HG (Figure 2).

The largest ROC AUC was shown by the relative expression of the *SLC26A7* gene (AUC = 0.816, very good model performance), followed by genes *TFF3* (AUC = 0.788) and *TPO* (AUC = 0.780) (good model performance for both genes).

The algorithm of plotting a decision tree (C4.5) belonging to the class of logical methods was used to stratify the cytology specimens into different groups. Figure 3 illustrates the resulting decision tree.

The decision tree involved the genes *SLC26A7* and *CPQ* as well as miR-21 and the ratio of mtDNA to nDNA. The tree has a linear structure, and basically, an FNA sample is classified as high-grade if the expression of *SLC26A7* is less than the cutoff set by the algorithm (0.1), miR-21 expression is less than a cutoff of 56.96, the mtDNA/nDNA ratio is greater than 630, and the relative *SLC26A7* expression is less than 0.257. If one of these conditions is not met, then the tissue sample is classified as non-high-grade. Data on diagnostic characteristics of this decision tree in terms of the identification of PTC and PTC HG are summarized in Table 3.

The resulting classifier based on the genes *SLC26A7* and *CPQ*, as well as miR-21 and mtDNA as markers, was found to be highly specific (100%) but insufficiently sensitive (54%). In comparison, if the *SLC26A7* gene was the only one used as a marker, then the sensitivity was the same (54%), but the specificity was lower (87.6%).

## 4. Discussion

In this study, we assessed the feasibility of preoperative detection of PTC HG (Bethesda V–VI) in cytology specimens using molecular markers. PTC is the most common malignant tumor of the thyroid gland (accounting for 85% of all thyroid cancer cases) and usually has an indolent course, with an overall 5-year survival rate of ~97.5%. Only a small percentage of well-differentiated carcinomas behave clinically aggressively and can lead to the death of the patient [46]. Some experts consider these carcinomas “true carcinomas” of the thyroid gland [47]. Until recently, however, researchers have not deemed it important to distinguish PTC HG as a separate subtype because this subdivision—according to the data available at the time—has not had any clinical or prognostic consequences [48]. According to more recent data, PTC HG should be distinguished as a separate subtype owing to its aggressive clinical course. Because molecular changes in PTC HG do not match the molecular changes in PDTC, it is appropriate to regard aggressive forms of PTC (e.g., PTC HG) as separate, not only from papillary carcinoma but also from PDTC [7].

Thus, it is crucial to search for efficient markers for the preoperative detection of PTC HG tumors. Such markers could be miRNAs, which are reported to be associated with an aggressive course of PTC [37]. We examined eight miRNAs in preoperative cytology specimens but failed to find any significant differences in miRNA expression between PTC and PTC HG.

Protein-coding genes responsible for various processes in thyrocytes are other candidates for molecular markers. Among the 14 studied genes, significant differences in expression levels between groups PTC and PTC HG were only revealed for genes *SLC26A7*, *TFF3*, and *TPO*. It should be noted that these three genes are related to the normal functioning of the thyroid gland rather than to malignant transformation. For instance, the *TPO* gene encodes thyroid peroxidase, which plays an important role in the biosynthesis of hormones; the *SLC26A7* gene codes for a transmembrane ion transporter, which apparently plays an important part in the maintenance of an optimal ionic environment and pH for activities of the enzymes (including TPO) involved in the biosynthesis of thyroid hormones. The *TFF3* gene is not specifically associated with the thyroid gland, although its expression is highest there (https://www.gtexportal.org/home/gene/TFF3 (accessed on 22 November 2024)); it encodes a protein belonging to the family of small secretory molecules participating in the maintenance and restoration of structural integrity of the epithelium [49]. Thus, it can be theorized that the differences between PTC HG and PTC are related to the downregulation of genes differentially expressed in the thyroid gland, i.e., with a loss of differentiation by the tumor.

Nonetheless, only one gene (*SLC26A7*) turned out to be crucial for detecting PTC HG. This simple classifier is characterized by the largest AUC ROC (0.816) in differentiation between the groups PTC and PTC HG and proved to be the key element of the decision tree by ensuring 54% sensitivity and 87.6% specificity. The other markers in the decision tree (*CPQ*, miR-21, and mtDNA) enhanced specificity (to 100%) but were possibly chosen for the algorithm randomly because of the distinctive features of the sampling procedure. On the contrary, the expression of the *SLC26A7* gene is nonrandomly associated with the development of PTC HG. SLC26A7 plays an important, though not fully understood, role in thyroid hormone biosynthesis and is expressed predominantly in the thyroid gland, with dysfunction of this gene resulting in congenital goitrous hypothyroidism [50]. There is evidence that SLC26A7 is another iodine transporter because it has appreciable homology to the *SLC26A4* gene (encoding an iodide exporter, also known as pendrin, which transports iodide from follicular cells to the lumen). At least when SLC26A4 is not functioning properly, SLC26A7 can perform iodine transport into the follicular lumen, thereby replacing SLC26A4 [51]. Just as in our study, *SLC26A7* expression has been found to be significantly downregulated in patients with anaplastic thyroid carcinoma [26]; moreover, according to ref. [39], in PTC, SLC26A7 downregulation correlates with an elevated risk of metastases outside the thyroid.

The accumulation of data, refinement of morphological criteria, and expansion of knowledge about thyroid tumor biology will help to identify more aggressive and life-threatening subtypes of cancer at an early stage. In the present study, six out of thirteen patients with PTC HG had early disease stages T1 and T2. Early preoperative diagnosis of PTC HG in patients with early stages of this cancer will allow clinicians to adjust the treatment plan toward a broader surgical extent and lymph node dissection, and it will possibly provide indications for subsequent radioactive iodine therapy.

This study has several limitations: only 13 tissue samples of PTC HG were analyzed in the work; in the future, of course, it will be necessary to increase the sample size. Also, only a small set of markers was evaluated, which means that other potential markers could have been overlooked. Another limitation is that this study deals with a specific group of Russian patients; this means that the findings cannot be extrapolated to patient groups in other geographic locations or to other ethnic groups, owing to the effects of environmental and genetic factors. Further research will require high-throughput methods to find a sufficient number of markers for the differential diagnosis of PTC HG.

## Figures and Tables

**Figure 1 diagnostics-14-02652-f001:**
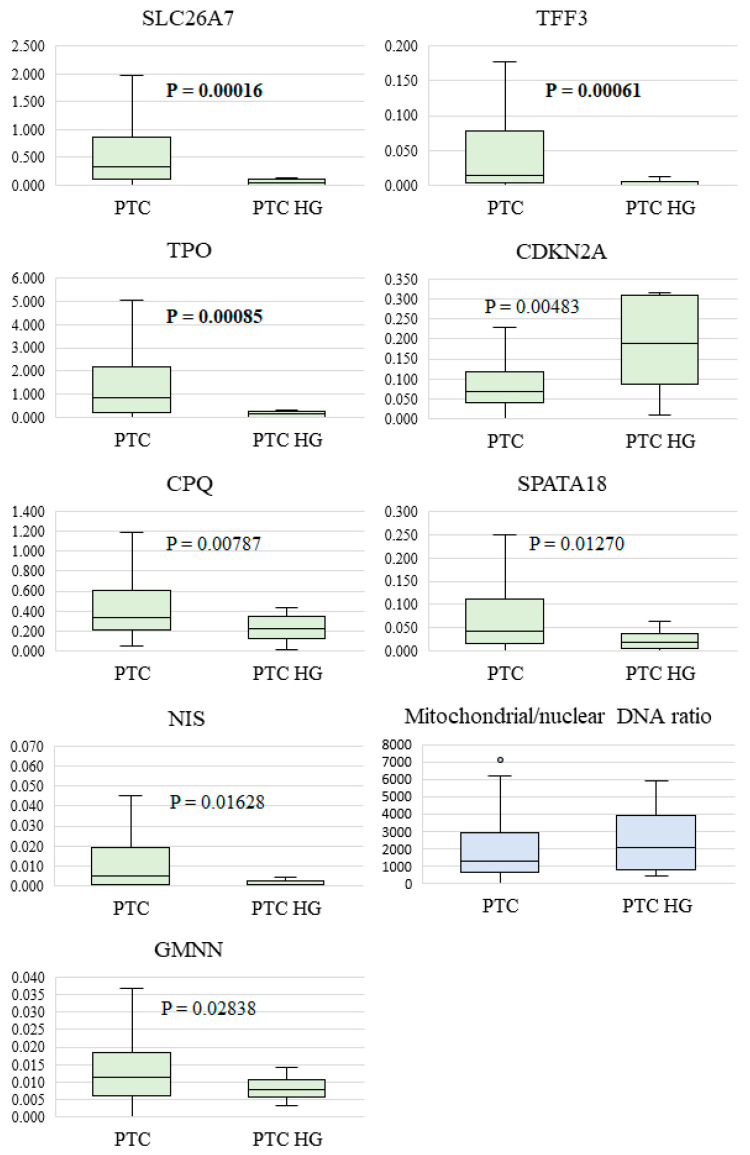
Relative expression levels of eight genes and the mtDNA/nDNA ratio in PTC and PTC HG specimens. The median, upper, and lower quartiles (box), a nonoutlier range (whiskers), and outliers (circles) are indicated. The *p*-values at the significance level chosen in this study (<0.0022) are shown in bold.

**Figure 2 diagnostics-14-02652-f002:**
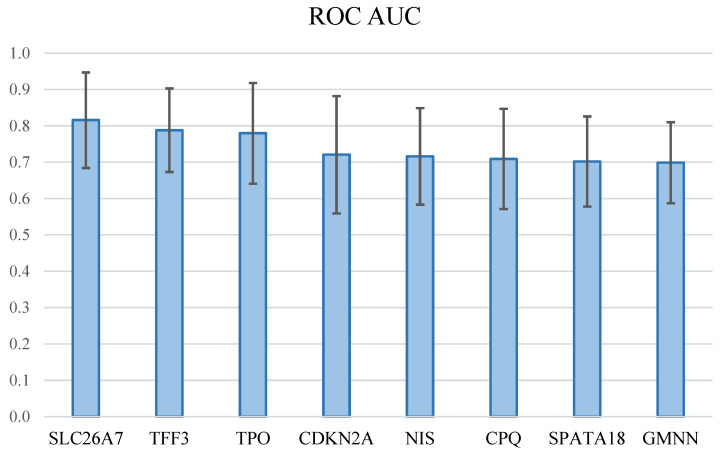
Areas under the ROC curves (ROC AUC) for eight mRNAs for assessing the distinguishability of groups PTC and PTC HG. Whiskers denote 95% confidence intervals.

**Figure 3 diagnostics-14-02652-f003:**
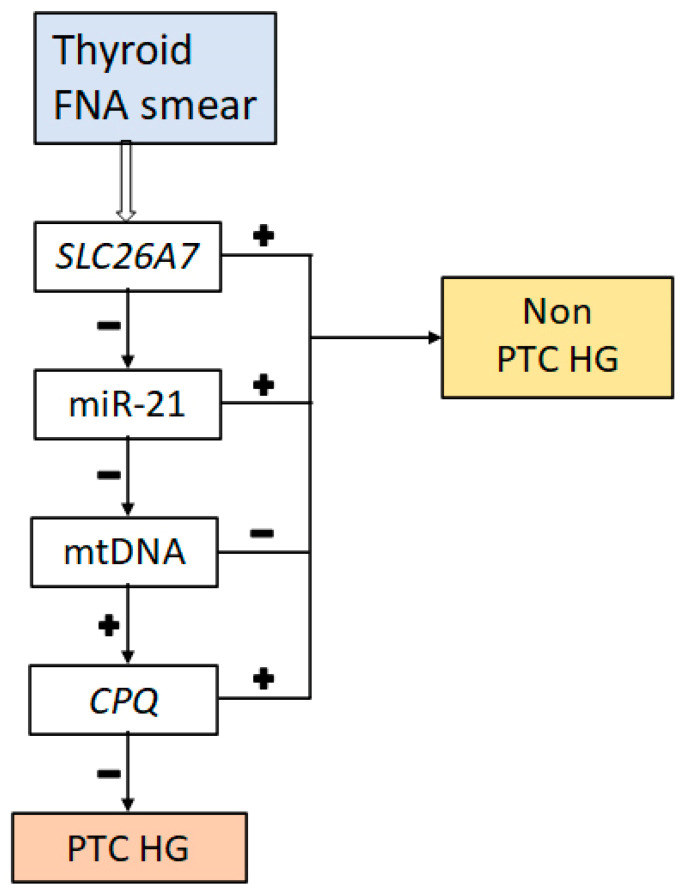
The decision tree for classifying samples into benign or malignant ones followed by cancer typing. + or −: exceeding the chosen cutoff or not exceeding it.

**Table 1 diagnostics-14-02652-t001:** Clinical characteristics of the patients.

Characteristic	Value, *n* (%)	
	PTC (*n* = 97)	PTC HG (*n* = 13)
Age, median (Q1-Q3)	47.5 (37–60.25)	64 (48.75–74.25)
Sex ratio (M/F)	21/76	3/10
Metastasis to central cervical lymph nodes N1a	26 (26.8%)	2 (15.4%)
Metastasis to lateral cervical lymph nodes N1b	19 (19.6%)	6 (46.2%)
Multifocality	58 (59.8%)	3 (23%)
Extrathyroidal extension (macroscopic invasion)	18 (18.6%)	7 (53.8%)
Distant metastasis	0	0

**Table 2 diagnostics-14-02652-t002:** Statistical significance of differences in the pairwise comparison of groups PTC (*n* = 97) and PTC HG (*n* = 13).

Marker	*p*-Value
miR-146b	0.452583
miR-199b	0.086416
miR-221	0.744162
miR-223	0.204396
miR-31	0.067463
miR-375	0.767790
miR-551b	0.629983
miR-21	0.359206
mtDNA	0.206242
*FN1*	0.142648
*GMNN*	0.028378
*CDKN2A*	0.004832
*TPO*	0.000850
*SLC26A7*	0.000166
*HMGA2*	0.558307
*CPQ*	0.007873
*SPATA18*	0.012699
*APOE*	0.104936
*DIO1*	0.929108
*NIS*	0.016281
*TFF3*	0.000614
*TMPRSS4*	0.547838
*TSHR*	0.458833

Significant differences are boldfaced (adjusted *p*-value < 0.0022).

**Table 3 diagnostics-14-02652-t003:** Diagnostic characteristics (including the 95% confidence interval) of PTC and PTC HG identification either by means of the decision tree or exclusively by means of the *SLC26A7* gene.

Diagnostic Characteristics	Decision Tree	*SLC26A7*
Sensitivity	53.8% (25.1–80.8%)	53.8% (25.1–80.8%)
Specificity	100.0% (96.3–100.0%)	87.6% (79.4–93.4%)
Accuracy	94.5% (88.5–97.8%)	83.6% (75.5–90.0%)
Positive predictive value	100.0% (59.0–100.0%)	36.8% (36.8–54.8%)
Negative predictive value	94.2% (90.0–96.7%)	93.4% (88.7–96.2%)

## Data Availability

The raw data supporting the conclusions of this article will be made available by the authors upon request.

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
