# Peer review of "Could SLC26A7 Be a Promising Marker for Preoperative Diagnosis of High-Grade Papillary Thyroid Carcinoma?"

_diagnostics, 2024, doi:10.3390/diagnostics14232652_

Round 1
Reviewer 1 Report
Comments and Suggestions for Authors
I would like to extend my congratulations to the authors for their excellent work. FNA is a commonly utilized diagnostic tool for PTC, and the identification of molecular markers is consistently essential. The occurrence of PTCHG in a history of PTC is relatively uncommon. Identifying distinctions in mRNA and miRNA will enhance the precision of the early diagnostic process.
Minor comments:
The manuscript is well-written, and the scientific language is clear and accessible. I recommend that you make the introduction section more original. In this case, the subject matter appears to be somewhat disorganized.
It is recommended that the authors first introduce the topic of PTC and its subtypes and then describe the processes performed according to the WHO classification system and FNA. This approach would facilitate a more coherent and fluid reading of the text. In summary, the initial five paragraphs can be organized logically.
A reference article should be included for the reference genes (mRNA and miRNA) selected in the RT-PCR stage.
The relatively small number of samples in the PTC HG group (n = 13) represents a limitation of the study. However, it should be noted that the researchers have invested significant time and effort over two years in collecting and processing these samples. In light of these considerations, it would be advisable to include a discussion of the study's limitations in an appropriate section of the text. Notwithstanding these limitations, the significant differences observed in TFF3, SLC26A7, and TPO are noteworthy. Furthermore, combining Figures 2 and 3 would enhance the clarity and accessibility of the presentation.
*Please include PMID:34593695 reference in your discussion section. This is the only missing reference for PTC and SLC26A7 expression.
I have a minor proposal for researchers. It is recommended that the researchers present the various molecular markers identified as statistically significant within the context of the study as a single figure, with different aspects delineated.
*GeneMania (see link: https://genemania.org/search/homo-sapiens/TFF3/SLC26A7/TPO).
In the attached file, two separate figures have been exported, and a summary has been created using approaches such as shared protein domain and co-expression. These databases could be used to enrich the discussion section and increase the number of possible citations.

Author Response
Thank you for the positive evaluation of our study.

Reviewer 2 Report
Comments and Suggestions for Authors
The authors conducted various gene and miRNA analyses in the article titled ‘SLC26A7 Is a Promising Marker for Preoperative Diagnosis of 2 High-Grade Papillary Thyroid Carcinoma’ to use a preoperative molecular marker for high-grade papillary thyroid cancer (PTC HG). The subject of the study is quite current and interesting. I think it would be useful to make some suggestions by mentioning the strengths and weaknesses of the study for further development.
Title and Abstract
The study title comprehensively reflects the content of the study. However, the diagnostic performance of markers such as SLC26A7 is limited by 54% sensitivity. This means that more than half of the PTC HG cases cannot be detected using these markers. This makes the statement ‘promising marker’ exaggerated and unfair. Therefore, I think the title could be changed to ‘Could SLC26A7 be a Promising Marker for Preoperative Diagnosis of High-Grade Papillary Thyroid Carcinoma?’ The abstract provides a general summary of the study and is comprehensive. However, adding 1-2 sentences to the abstract emphasizing the importance of the study provides information about the purpose of the study.
Introduction
The introduction section summarizes the current literature on how high-grade papillary thyroid cancer differs from other thyroid cancer subtypes.
The aim of the study was stated as finding reliable molecular markers to preoperatively diagnose PTC HG. However, explaining the place and advantages of such markers in clinical applications in more detail can better emphasize the purpose of the study.
Methods
The methods used are clearly stated, and the procedures are detailed in the Material and Method section; however, the details of RNA extraction and the validation process of RT-PCR protocols should be explained in more detail. A limited list of selected gene and miRNA markers were tested in the study. The association of only certain genes with different expression levels means that other potential markers may have been overlooked. In addition, the inclusion and exclusion criteria of the study participants should be detailed and even presented in a tabular form. Statistical analysis is explained sufficiently and understandably.
Results
The Results section is quite structured, and the markers showing significant differences are presented in detail. In particular, it emphasized the differences between the PTC and PTC HG groups in the SLC26A7, TFF3, and TPO genes. These findings are explained in the context of the biological function of each gene and its relationship with the undifferentiated PTC HG. Although this section should be more in the discussion section, I think that mentioning it immediately after the results will increase the reader's understanding.
The diagnostic performance of markers such as SLC26A7 is limited to 54% sensitivity. This means that more than half of the PTC HG cases cannot be detected using these markers. Considering the need for high sensitivity, especially in preoperative diagnosis, more powerful markers or combinations should be evaluated.
In addition, the markers examined in the study were not compared with other existing diagnostic methods. For example, comparisons with widely used diagnostic panels such as ThyGenX or ThyraMIR should be made to further study the advantages and disadvantages of SLC26A7 and other markers compared to existing tests.
The graphs and tables visually support the results of the study. However, the algorithmic structure of the decision tree needs to be explained in more detail in the text.
Discussion
The discussion section relates the findings to the existing literature and explains the potential importance of the study results in clinical practice. In particular, the changes that preoperative diagnosis of PTC HG may create in treatment planning are emphasized.
However, the limitations of the study are not addressed in this section. The most important limitation of the study is the very small sample size.
While 97 PTC patients provide a sufficient sample size, the fact that only 13 PTCHG cases constitutes a major limitation of the study. How do the authors intend to overcome this? Another limitation is that the study focused on a specific patient group in Russia. This means that the results cannot be directly generalized to patient groups in other geographic regions or different ethnic populations due to the effects of genetic and environmental factors.
I wish the authors success in their study.
Comments on the Quality of English Language
The English language contains many grammatical and typo errors and should be revised throughout the paper.
Author Response
Thank you very much for your time and your interest in our manuscript.

Round 2
Reviewer 2 Report
Comments and Suggestions for Authors
The authors have responded satisfactorily to the comments and made substantial revisions. I wish the authors success in their work.